# Cold-Adapted Glutathione S-Transferases from Antarctic Psychrophilic Bacterium *Halomonas* sp. ANT108: Heterologous Expression, Characterization, and Oxidative Resistance

**DOI:** 10.3390/md17030147

**Published:** 2019-03-01

**Authors:** Yanhua Hou, Chenhui Qiao, Yifan Wang, Yatong Wang, Xiulian Ren, Qifeng Wei, Quanfu Wang

**Affiliations:** School of Marine Science and Technology, Harbin Institute of Technology, Weihai 264209, China; marry7718@163.com (Y.H.); mariahit@163.com (C.Q.); daid01@126.com (Y.W.); wangyatong199311@163.com (Y.W.); renxiulian@126.com (X.R.); weiqifeng163@163.com (Q.W.)

**Keywords:** glutathione S-transferases, cold-adapted, Antarctic, antioxidant defense, homology modeling

## Abstract

Glutathione S-transferases are one of the most important antioxidant enzymes to protect against oxidative damage induced by reactive oxygen species. In this study, a novel *gst* gene, designated as *hsgst*, was derived from Antarctic sea ice bacterium *Halomonas* sp. ANT108 and expressed in *Escherichia coli* (*E. coli*) BL21. The *hsgst* gene was 603 bp in length and encoded a protein of 200 amino acids. Compared with the mesophilic EcGST, homology modeling indicated HsGST had some structural characteristics of cold-adapted enzymes, such as higher frequency of glycine residues, lower frequency of proline and arginine residues, and reduced electrostatic interactions, which might be in relation to the high catalytic efficiency at low temperature. The recombinant HsGST (rHsGST) was purified to apparent homogeneity with Ni-affinity chromatography and its biochemical properties were investigated. The specific activity of the purified rHsGST was 254.20 nmol/min/mg. The optimum temperature and pH of enzyme were 25 °C and 7.5, respectively. Most importantly, rHsGST retained 41.67% of its maximal activity at 0 °C. 2.0 M NaCl and 0.2% H_2_O_2_ had no effect on the enzyme activity. Moreover, rHsGST exhibited its protective effects against oxidative stresses in *E. coli* cells. Due to its high catalytic efficiency and oxidative resistance at low temperature, rHsGST may be a potential candidate as antioxidant in low temperature health foods.

## 1. Introduction

Antarctica is isolated geographically from other continents and presents an extremely harsh environment on the earth. Antarctic environment has the characteristics of low temperature, high salinity, low nutrient availability, and strong ultraviolet (UV) radiation [1]. To adapt to the extreme conditions, microorganisms can produce a series of enzymes and display unique metabolic properties. Therefore, Antarctic microorganisms play a crucial part in the field of biomass conversion and have great potential applications, which have aroused more attention [1]. Sea ice is an important component of Antarctic climate and ecosystems. The increased oxygen concentration associated with the high salt concentration and low temperature of sea ice can induce the generation of reactive oxygen species (ROS) [2]. To remove the influence of ROS, sea ice microorganisms have especial antioxidant defense systems to protect against oxidative damage [2]. In the antioxidant defense systems, Glutathione S-transferases (GSTs) play a vital role in the regulation of the detoxification and redox balance of ROS [3]. Additionally, GSTs can catalyze the combination of glutathione (GSH) and various active electrophiles through affinity attacks and couplings [4]. GSTs have been widely described from animals [5], plants [6], and microorganisms [7]. Recently, a GST with an optimum temperature of 40 °C has been identified in the Antarctic psychrophilic bacteria *Pseudoalteromonas* sp. [8]. The crystallization and X-ray crystallographic of another GST from Antarctic clam *Laternula elliptica* were also studied [9]. To date, little information is available concerning the structural characteristics and biochemical properties of GSTs from Antarctic organisms.

Soluble GSTs can be divided into three categories: cytosolic, microsomal, and mitochondrial GSTs [10]. GSTs are composed of an amino acid chain, which contains the C-terminal domains and N-terminal domains (like-thioredoxin) [11]. The N-terminal domains with α-helixes and β-strands can contribute to GSH binding sites, and the C-terminal domains can provide more amino acid residues that interact with a variety of hydrophobic xenobiotics substrates. Besides, compared with N-terminal domains, the C-terminals of GSTs exhibit more structural changes, which can recognize and bind to a variety of known electrophilic compounds as GST substrates [12]. Currently, GSTs have been widely used as biomarkers of toxicity [13] and in pharmaceutical industries to scavenge organic hydroperoxides and ROS [14]. Furthermore, GSTs have been used as an indicator of certain diseases, especially hepatic infection [15]. Therefore, GSTs have different biological activities due to the differences of the conserved sequence regions, substrate specificity, and biochemical characterizations.

Antarctic sea ice microorganisms would be a novel source of antioxidant enzymes. Based on our recent studies, several cold-active antioxidant proteins were separated from Antarctic sea ice microorganisms and widely used in industrial sectors due to their excellent biochemical properties [16,17]. *Halomonas* is the type genus of the family *Halomonadaceae* belonging to the class *Gammaproteobacteria.* The genome sequence of *Halomonas* sp. strain KO116 isolated from sea surface [18] and the physiological features of *Halomonas lionensis* from sea sediment have also been reported [19]. In this study, a novel *gst* gene (*hsgst*) was cloned from Antarctic sea ice bacterium *Halomonas* sp. ANT108 and heterologous expressed in *E. coli* BL21. Furthermore, its biochemical properties and oxidative resistance were investigated.

## 2. Results and Discussion

### 2.1. Identification of the hsgst Gene

Sequence analysis of the full-length *hsgst* gene revealed an open reading frame (ORF) of 603 bp, encoding a protein of 200 amino acids with a predicted molecular weight of 21.76 kDa and the theoretical isoelectric point (pI) of 5.79. The length of *Taenia multiceps gst* gene was 606 bp encoding 201 amino acids, which was close to the length of *hsgst* gene [20]. Besides, the length of *gst* gene from *Cydia pomonella* and *Sus scrofa* were 648 bp encoding 215 amino acids (24.2 kDa) and 669 bp encoding 222 amino acids (25.3 kDa), respectively, which were both slightly longer than the *hsgst* gene [6]. The nucleotide sequence of the *hsgst* dene was deposited into GenBank database (accession numbers MH719093).

Based on sequence alignments with the related GSTs, HsGST shared the highest identity (90.0%) with HcGST from *Halomonas campaniensis*. The dimer interface of HsGST was comprised of L^91^, A^92^, G^95^, L^96^, G^99^ and R^133^ (Figure 1). In general, soluble GSTs have two binding domains with the GSH binding site (G-site) and the substrate binding site (H-site) [21]. The H-site (Y^8^, F^9^, V^11^, R^14^, V^111^, Y^115^, N^211^ and G^212^) and G-site (Y^8^, R^14^, W^42^, K^50^, Q^57^, L^58^, P^59^, Q^71^, S^72^, E^104^ and D^105^) were identified in the *gst* gene from *Laternula elliptica* [22]. Similarly, VpGSTp from *Venerupis philippinarum* also possessed the H-site (Y^8^, F^9^, V^11^, R^14^, V^103^, Y^107^, N^200^ and G^201^) and G-site (Y^8^, R^14^, W^39^, K^45^, Q^52^, L^53^, P^54^, Q^65^, S^66^, E^96^ and D^97^) [23]. However, in this study, only one H-site (Q^98^, M^101^, D^102^, A^161^ and Y^164^) was found in the sequence of HsGST. Additionally, the N-terminal domain interface (L^91^, Q^98^, L^153^, I^156^, T^157^, V^160^ and Y^164^) of HsGST was also identified in this sequence (Figure 1).

### 2.2. Homology Modeling Analysis

The 3D structure of HsGST showed that it consisted of 8 α-helixes and 4 β-strands (Figure 2A). For the validation of the structural model, 82.00% of the residues had an averaged 3D-1D score ≥ 0.2. The ERRAT program also showed an overall quality value of 94.27% for the 3D structure. These parameters demonstrated that the structural model of HsGST was well qualified. As can be seen in Figure 2B, the structural model of HsGST superimposed well with EcGST. The H-sites of HsGST and EcGST were also effectively located in close proximity (Figure 2B). EcGST (PDB ID: 3R2Q) was a mesophilic GST from *E. coli* K12 encoded 202 amino acids, and the sequence identity of EcGST to HsGST was 30% using BLASTp. In fact, there were differences in the frequencies of the amino acid residues between the cold-adapted enzymes and the homologous mesophilic enzymes [24]. As shown in Table 1, in the comparison with EcGST, HsGST possessed of a higher frequency of G (Gly), lower frequency of P (Pro) and R (Arg), regarded as a structural characteristic of cold-adapted enzymes, which endowed enzymes with increased conformational flexibility and low-temperature catalytic competence [24].

Furthermore, electrostatic interactions were an important factor for maintaining the secondary and tertiary structure in cold-adapted enzymes [25]. Compared to EcGST, HsGST exhibited reduced electrostatic interactions, especially hydrogen bonds and salt bridges, resulting in a decrease in the rigidity of the proteins, which were also obviously observed in other cold-adapted enzymes such as dienelactone hydrolase [26] and glycosylase [27]. In addition, in the comparison with EcGST, HsGST had less hydrophobic interactions, which might contribute to the increased structural flexibility and thermolability of cold-adapted enzymes [26,28]. The increased structural flexibility could improve the efficiency of substrates binding to the catalytic sites with high possibility, thereby reducing activation energy and increasing substrates turnover rates [26]. 

### 2.3. Expression and Purification of rHsGST

As shown in Figure 3 lane 3, the recombinant protein with an estimated molecular mass of 28 kDa was expressed in *E. coli* BL21. The purified protein using nickel nitrilotriacetic acid (Ni-NTA) resins exhibited a single band in terms of sodium dodecyl sulfate polyacrylamide gel electrophoresis (SDS-PAGE) (Figure 3, lane 4). In addition, 2.18 mg protein of rHsGST was purified from the crude extract. The purification fold and yields of rHsGST were 3.62 and 27.75%, respectively. The specific activity of rHsGST was 254.20 nmol/min/mg, which was higher than the GST isolated from *Liposcelis entomophila* (183 nmol/min/mg) and lower than that isolated from *L. bostrychophila* (412 nmol/min/mg) [29].

### 2.4. Biochemical Characterizations of rHsGST

The optimum temperature of rHsGST was measured over a temperature range of 0–50 °C (Figure 4A). rHsGST had the maximal activity at 25 °C and remained 41.67% of the maximal activity even at 0 °C, while the enzyme was inactivated at 50 °C. However, the optimum temperature of the GST from the *Laternula elliptica* [22] and GST from codling moth (*Cydia pomonella*) [30] were 35 °C and 50 °C, respectively. For the thermostability of rHsGST, the half-life of the enzyme was at 35 °C for 90 min, and rHsGST was completely inactivated after 1 h incubation at 45 °C (Figure 4B). According to other reports, GSTs from *Pseudoalteromonas* sp. [8] and *Monopterus albus* [31] retained 50% and 90% of the maximal activity at 40 °C for 15 min, respectively. These results indicated that rHsGST belonged to the cold-adapted enzyme, which was consistent with the results of homology modeling. rHsGST showed the maximal activity at pH 7.5 (Figure 4C), while the optimum pH of GST from *Spodoptera exigua* [32] and *Locusta migratoria manilensis* [33] was 6.0 and 8.0, respectively. Moreover, rHsGST retained about 50% and 65% of its maximal activity at pH 6.5 and pH 8.5, respectively. Similar results were also obtained in rRfGSTθ from black rockfish [34]. Consequently, it possessed good activity in neutral and weakly alkaline conditions. The stability of rHsGST was studied at various pH values from 5.0 to 9.0 (Figure 4D). The enzyme retained more than 60% of the maximal activity at pH 5.0–8.5. It illustrated that rHsGST had better stability over a wide pH range, which was similar with the GST from the locust [33]. 

As shown in Figure 5, rHsGST exhibited more than 80% of its initial activity at 0.5–2 M NaCl. However, the activity declined sharply at 2.5 M. These results illustrated that the rHsGST was a salt-tolerant enzyme that could adapt to the Antarctic sea ice environment with a high salt concentration. The activity of GST isolated from *Pseudoalteromonas* sp. ANT506 was lower than rHsGST, which remained about 50% of the initial activity in the existence of 2.0 M NaCl [8].

The effects of different reagents on rHsGST activity were also investigated (Table 2). The activity of rHsGST was not detected in the presence of metal ions Ca^2+^, Ni^2+^, Cu^2+^ and Sn^2+^. On the contrary, the activity of GST from *Trichinella spiralis* was promoted by Ni^2+^ [35]. Interestingly, Fe^2+^ could improve the rHsGST activity, and H_2_O_2_ also kept the enzyme activity with 97.09%. rHsGST that was treated with ethanol showed the strongest inhibitory effect, and the activity was only 29.03% of control. Besides, the activity of rHsGST treated with SDS retained 54.84%, a similar result was also obtained in the recent study of the TsGST from *Trichinella spiralis* [35]. 

### 2.5. Kinetics and Thermodynamics Parameters

Kinetic parameters were determined by linweaver-burk plot, *V*_max_ of substrate chlorodinitrobenzene (CDNB) and GSH were 714.29 and 243.90 nmol/min/mg, respectively (Table 3). Compared with the results of LmGSTu1, the *V*_max_ of CDNB in this study was much higher than 250 nmol/min/mg [36]. Furthermore, the *K*_m_ of GSH was 0.27 mM, which was lower than the value of CDNB (2.86 mM). These consequences illustrated that rHsGST had a higher substrate affinity for GSH than for CDNB.

Thermodynamic parameters were investigated using the CDNB as the substrate (Table 4). Obviously, as the temperature increased from 10 °C to 30 °C, the values of Δ*G* appeared a growing tendency. The values of Δ*H* were changed from 34.11 to 33.65 KJ/mol at temperatures from 10 to 30 °C. Besides, the values of Δ*S* were −101.48, −101.63, −101.77, −101.91, and −102.05 J/mol, followed by the temperature 10, 15, 20, 25, and 30 °C, respectively. The negative value of Δ*S* could increase the order of the activated transition state of catalysis [37].

### 2.6. Disk Diffusion Assay

It is known that H_2_O_2_ can cause oxidative stress that may induce cell dysfunction [38]. As can be seen from Figure 6, the clearance zones in the recombinant bacteria plate were significantly smaller than those of controls (*p* < 0.05). This result indicated that rHsGST had an important antioxidant effect on helping living tissues to overcome the oxidative stress caused by H_2_O_2_. Similarly, GSTs from *Haliotis discus discus* and *Apis cerana cerana* could also protect cells against oxidative stress produced by H_2_O_2_ [39,40].

## 3. Materials and Methods

### 3.1. Strains and Materials

Antarctic sea ice bacterial samples (68° 30′ E, 65° 00′ S) were cultured in 2216E medium at the logarithmic growth phase, then the culture solution was extracted and streaked into a solid medium by inoculating loop. The medium was then cultured at 12 °C for 72 h to pick up a single colony, which was identified as *Halomonas* sp. according to the sequence analysis of 16S rRNA genes (accession number: MK494178). The isolated strain had an optimum growth temperature of 10–12 °C. Plasmid vector pET-28a(+) and receptor *E. coli* BL21 were maintained by our laboratory. T_4_ DNA ligase, *Bam*HI and *Hin*dIII and SanPrep column DNA gel recovery kit were purchased from Takara Biotechnology (Dalian) Co., Ltd (Beijing, China).

### 3.2. Identification of hsgst Gene 

Using high-throughput technologies to sequence and annotate the genome of *Halomonas* sp. ANT108 (data not shown). The sequencing of full-length *hsgst* gene was amplified by PCR applying by the forward primers 5′-ATAGGATCCATGCAGC TCTATTTAA-3′ and the reverse primer 5′-AGTAAGCTTGTTCAAACGTGG TAAG-3′ (the *Bam*HI and *Hin*dIII sites were underlined, respectively) based on its genome sequence. The complete amino acid sequences of the HsGST were obtained through an ORF finder (http://www.ncbi.nlm.nih.gov/gorf/gorf.html). Multiple sequence alignments were performed using the Bioedit program. 

### 3.3. Analysis of HsGST

The 3D structure model of HsGST was established using the SWISS-MODEL server and verified using SAVES v5.0 (http://servicesn.mbi.ucla.edu/SAVES/) (University of California, Los Angeles, CA, USA). The visualization of homology modeling was performed using PyMOL software (Version No.2.2.0, DeLano Scientific LLC, CA, USA). Meanwhile, salt bridges were predicted to use ESBRI (Evaluating the Salt Bridges in Proteins). Besides, protein intramolecular interactions were predicted by the Protein Interactions Calculator (PIC) Online website (http://pic.mbu.iisc. ernet.in/job.html). 

### 3.4. Expression and Purification of hsgst Gene in E. coli

The *hsgst* gene was inserted into the plasmid of the pET-28a(+) vector. Then, the recombinant plasmid was transferred into the receptor *E. coli* BL21. The positive strain was verified by PCR and inoculated in LB medium supplemented with 0.1 mg/mL kanamycin (Kana). Afterwards, 1 mM isopropyl-β-d-thiogalactoside (IPTG) was added to induce the *hsgst* gene expression, and the 1 L cultures were pro-longed for 8 h at 25 °C. The induced cells were centrifuged and disrupted by sonication (JY96-IIN, Shanghai, China) to obtain inclusion bodies. Then the inclusion bodies were washed with PBS (pH 8.0) and treated with 8 M urea at 25 °C for 1 h. The mixture was centrifuged at 7500 rpm for 15 min. The supernatant was diluted for 30 times by adding PBS (pH 8.0) at 25 °C for 2 h, which was the procedure of the rHsGST refolding. The protein solution was centrifuged at 12,000 rpm for 15 min. The supernatant was the crude extract of rHsGST. The protein was purified using Ni-NTA resins affinity chromatography (GE Healthcare, Uppsala, Sweden). The protein molecular weight and purity of the rHsGST were determined by SDS-PAGE with 12.0% polyacrylamide gels.

### 3.5. Assay of rHsGST Activity and Protein Concentration 

GST activity was determined according to the conjugation of CDNB and GSH in the presence of the enzyme [12]. One unit of enzyme activity was defined as the quantity of CDNB conjugated product catalyzed synthetically per milligram per minute at 25 °C. Protein concentrations were measured by the method of Bradford [41]. 

### 3.6. Biochemical Characteristics of rHsGST 

The optimum temperature of the rHsGST was determined at different temperatures (0–50 °C) by the standard enzyme assay. To evaluate the thermostability of rHsGST, the enzyme was incubated at 35, 40, and 45 °C for 90 min, respectively. The optimum pH the rHsGST was measured at 25 °C and the following buffers were 0.1 mol/L acetate buffer (pH 5.0–5.8), 0.1 mol/L Na_2_HPO_4_/Na_2_HPO_4_ (pH 5.8–8.0), 0.1 mol/L Tris-HCl (pH 8.0–9.0). The stability of rHsGST at different pH was determined by incubating the enzyme in the buffers mentioned above at 25 °C for 1 h and the remaining activity was measured. Furthermore, the effect of NaCl on the rHsGST was evaluated by adding NaCl (0–3.0 M) and incubated at 25 °C for 1 h. The remaining activity was measured by means of the standard enzyme assay. In order to study the effects of various reagents on the rHsGST activity, the protein solution and different reagents solution were firstly mixed for 1 h at 25 °C, and then the remaining activity was measured by means of the standard enzyme assays.

### 3.7. Kinetics and Thermodynamics Parameters

Kinetic parameters were determined using five different concentrations of CDNB (0.5, 1.0, 1.5, 2.0, 2.5, 3.0, 3.5, 4.0 and 4.5 mM) and GSH (0.1, 0.2, 0.3, 0.4, 0.5, 0.6, 0.7, 0.8 and 0.9 mM) at 25 °C, respectively. *K*_m_ and *V*_max_ were measured using the Micchaelis-Menten equation [42]. Through the determination of kinetic, the *k*_cat_ was obtained. The *k*_cat_ was determined at 10–30 °C and the line was defined with ln*k*_cat_ and 1/T. Then, the Δ*G*, Δ*H*, and Δ*S* were calculated, respectively [42].

### 3.8. Disk Diffusion Assay

To compare the survival efficiency of untransformed BL21, BL21/pET-28a(+) and recombinant BL21/pET-28a(+)-HsGST, a disk diffusion assay was performed as described previously [43]. The IPTG-induced bacterial cultures were uniformly spread on LB agar plates, and two filter paper disks (diameter 3 mm) were placed equidistant on each agar plate. Then, 1.5 and 3 μL of 30% H_2_O_2_ were added to the filter paper disks, respectively. The treated plates were incubated at 37 °C for 12 h, and the diameters of the clearance zone were measured.

## 4. Conclusions

A novel *hsgst* gene from *Halomonas* sp. ANT108 was cloned, expressed, and characterized in the present study. Furthermore, HsGST had the structural characteristics of cold-adapted enzymes by homology modeling. After purification, rHsGST exhibited different catalytic capabilities compared with other GSTs, such as the optimum temperature, thermolability, and high tolerance in the presence of H_2_O_2_ and high salt concentration. Moreover, *E. coli* cells overexpressing *hsgst* displayed the protective effects against oxidative stress. Although there are already numerous GSTs available in the market, the high catalytic efficiency and oxidative resistance at low temperature of rHsGST may make it a potential candidate as an antioxidant in low temperature health foods such as marine surimi, protamine, and low temperature meat products.

## Figures and Tables

**Figure 1 marinedrugs-17-00147-f001:**
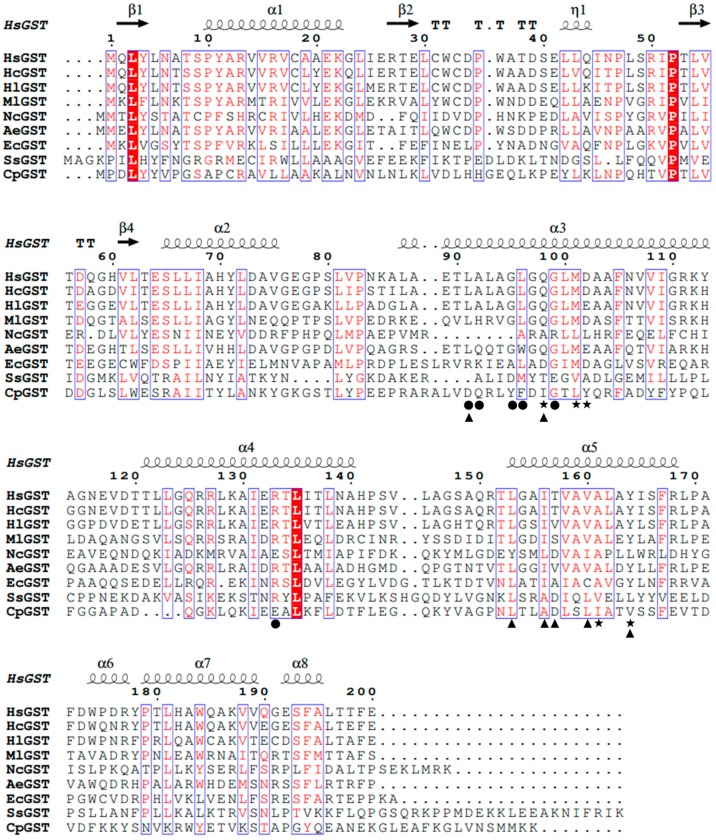
Alignment of deduced amino acid sequences of HsGST with other GSTs. HsGST, *Halomonas* sp. ANT108 GST (GI: MH719093); HcGST, *Halomonas campaniensis* GST (GI: WP_088700156); HlGST, *Halomonas lionensis* GST (GI: WP_083025363); MlGST, *Marinobacterium litorale* GST (GI: WP_027854519); AeGST, *Alkalilimnicola ehrlichii* GST (GI: WP_011629671); NcGST, *Nitrosomonas communis* GST (GI: AKH37377); EcGST, *E. coli* K12 GST (PDB ID: 3R2Q); SsGST, *Sus scrofa* GST (GI: NM214389) and CpGST, *Cydia pomonella* GST (GI: EU887533). Symbols: ●, dimer interface; ★, substrate binding pocket (H-site); ▲, N-terminal domain interface.

**Figure 2 marinedrugs-17-00147-f002:**
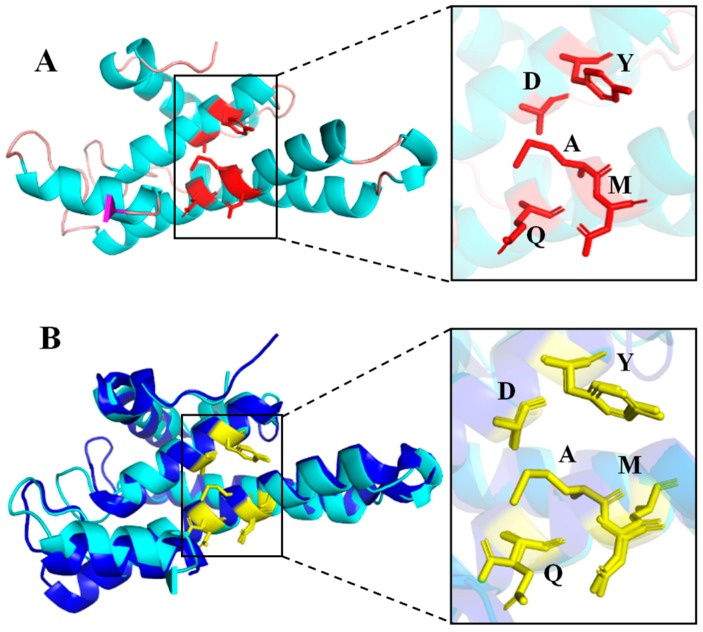
3D structure model of HsGST and structural superimposition with EcGST. (**A**) Cartoon representation of HsGST. The α-helices and β-strands are colored in blue and magenta, respectively. The catalytic triad residues are indicated as stick models colored in red. (**B**) The structural superposition of HsGST (cyan), EcGST (blue). The catalytic triad residues are indicated as stick models colored in yellow.

**Figure 3 marinedrugs-17-00147-f003:**
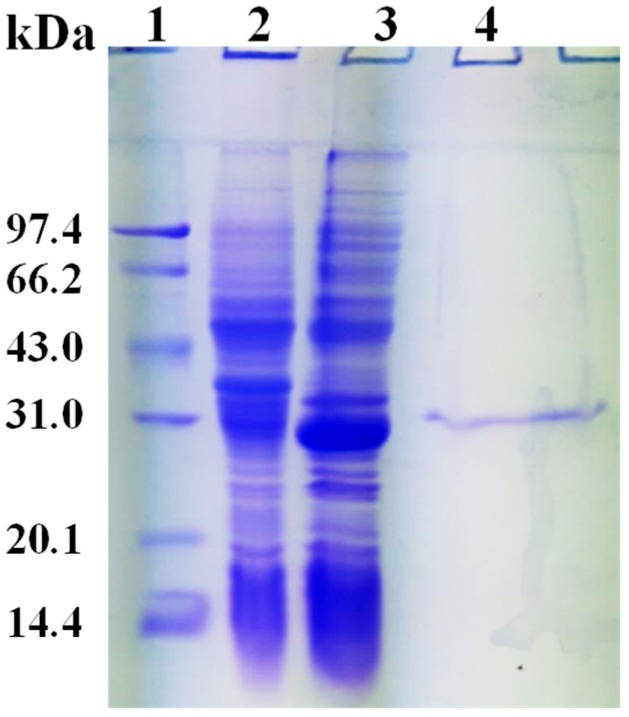
Expression and purification analysis of HsGST. Lane 1: protein molecular weight marker; lane 2: crude extract from the BL21/pET-28a(+); lane 3: crude extract from the BL21/pET-28a(+)-HsGST with IPTG induction; lane 4: purified rHsGST with Ni-NTA column.

**Figure 4 marinedrugs-17-00147-f004:**
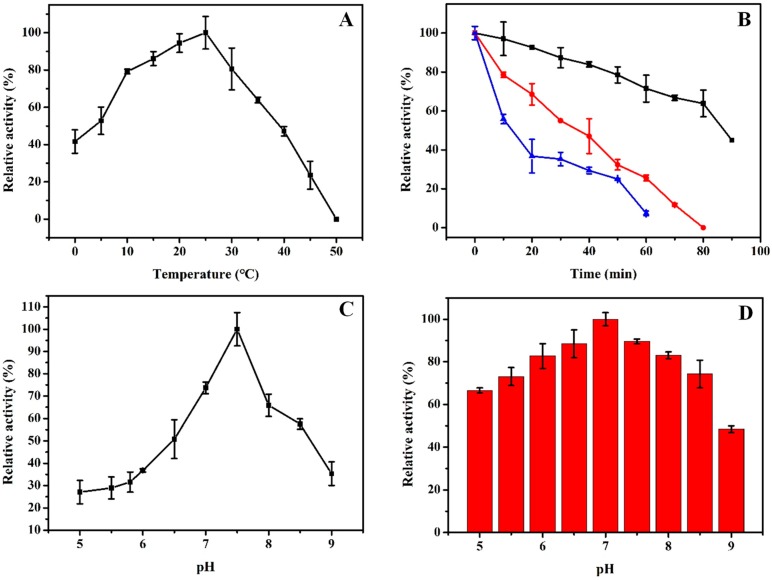
Effects of temperatures and pH on the purified rHsGST activity. (**A**) The optimal temperature was determined by measuring the activity at temperatures from 0 to 50 °C. (**B**) Effect of temperatures on the stability of the purified rHsGST. The enzyme was incubated at 35 °C (■, black), 40 °C (●, red), and 45 °C (▲, blue) for 90 min. (**C**) The optimal pH was determined by measuring the activity at pH from 5.0 to 9.0. (**D**) Effect of pH on the stability of the purified rHsGST. The enzyme was incubated at 25 °C for 1 h. The maximal activity was taken as 100%.

**Figure 5 marinedrugs-17-00147-f005:**
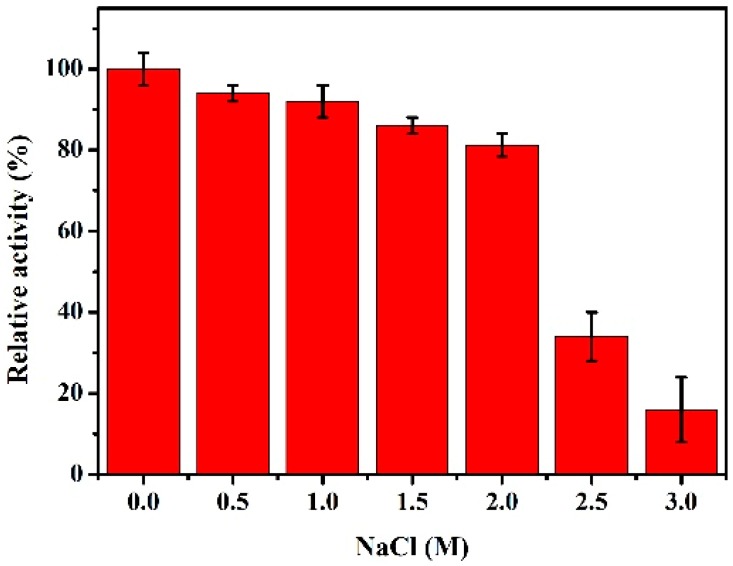
Effect of NaCl on the purified rHsGST activity.

**Figure 6 marinedrugs-17-00147-f006:**
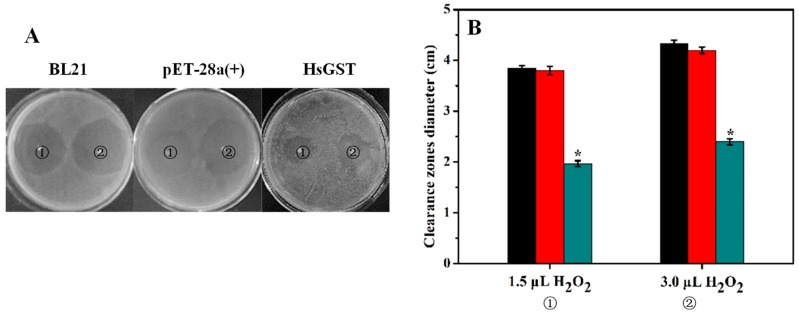
Disk diffusion assay against *E. coli* BL21, BL21/pET-28a(+) and BL21/pET-28a(+)-HsGST. (**A**) The clearance zone diameters (cm) were measured in the plates with *E. coli* BL21, BL21/pET-28a(+) and BL21/pET-28a(+)-HsGST after oxidative stress with H_2_O_2_. (**B**) The clearance zone diameters of BL21 (black), BL21/pET-28a(+) (red) and BL21/pET-28a(+)-HsGST (blue) were displayed in a bar graph form. Data are presented as mean (*n* = 3) ± SD. * *p* < 0.05, representing a significant difference from the control.

**Table 1 marinedrugs-17-00147-t001:** Comparison of structural adaptation features between HsGST and EcGST.

	HsGST	EcGST	Expected Effect on HsGST
Electrostatic interactions			Protein stability
Salt bridges	1	7	
Hydrogen bonds	165	272	
Aromatic interactions	7	6	
Cation-Pi interactions	5	4	
Hydrophobic interactions	159	180	Thermolability
G (Gly)	13	11	Flexibility
P (Pro)	10	13	
R (Arg)	11	14	
G substitution (HsGST→EcGST)	G76→N74, G78→A76, G95→A95, G97→A97, G110→Q111, G115→A116, G124→R125, G147→K147, G154→A154, G191→R191		
P substitution (EcGST→HsGST)	P64→L66, P84→A86, P114→Y113, P172→F172, P199→F199, P200→E200		
P substitution (HsGST→EcGST)	P142→V142, P170→V170, P175→C175		Stability

**Table 2 marinedrugs-17-00147-t002:** Effects of different reagents on the rHsGST activity.

Reagent	Conc	Relative Activity (%)	Reagent	Conc	Relative Activity (%)
None		100.00	Ba^2+^	5 mM	93.54 ± 2.90
K^+^	5 mM	38.71 ± 3.22	Ca^2+^	5 mM	ND
Ni^2+^	5 mM	ND	Mn^2+^	5 mM	70.97 ± 3.54
Fe^2+^	5 mM	148.38 ± 6.45	Ethanol	25%	29.03 ± 7.41
Zn^2+^	5 mM	41.94 ± 1.61	H_2_O_2_	0.2%	97.09 ± 2.90
Mg^2+^	5 mM	35.48 ± 7.41	SDS	5 mM	54.84 ± 6.45
Cu^2+^	5 mM	ND	EDTA	5 mM	56.45 ± 4.83
Sn^2+^	5 mM	ND	DTT	5 mM	ND

Conc: Concentration; ND: activity was not detected.

**Table 3 marinedrugs-17-00147-t003:** Kinetic constants of the rHsGST.

Substrate	CDNB	GSH
*V*_max_ (nmol/min/mg)	714.29	243.90
*K*_m_ (mM)	2.86	0.27
*k*_cat_ (1/s)	53.62	20.14
*k*_cat_/*K*_m_ (1/s/mM)	18.75	74.59

**Table 4 marinedrugs-17-00147-t004:** Thermodynamic constants of the rHsGST.

Temperature	10 °C	15 °C	20 °C	25 °C	30 °C
Δ*G* (KJ/mol)	62.85	63.35	63.82	64.15	64.59
Δ*H* (KJ/mol)	34.11	34.05	33.99	33.77	33.65
Δ*S* (J/mol)	−101.48	−101.63	−101.77	−101.91	−102.05
T × Δ*S* (KJ/mol)	−28.73	−29.29	−29.83	−30.38	−30.94

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
