# Peer review of "Cold-Adapted Glutathione S-Transferases from Antarctic Psychrophilic Bacterium Halomonas sp. ANT108: Heterologous Expression, Characterization, and Oxidative Resistance"

_marinedrugs, 2019, doi:10.3390/md17030147_

Round 1

Reviewer 1 Report

Basically, this manuscript provides some fundamental data with novelty. However, there are some concerns needed to be addressed before publication.

1.In Figure 3, I recommend the authors to combine Figure 3A and Figure 3B to on graph.

2.In Figure 3B lane 4, the band for rHsGST is not obvious. In addition, the authors did not explain the molecular weight and purity of rHsGST in the text.

3.The authors did not mention how to refold rHsGST.

4.In lines 163-164, this sentence seems not related to the topic discussed.

5.In Figure 6, the authors overexpressed HsGST in BL21. Generally, the overexpressed proteins within E. coli are in the form of inclusion bodies and do not refold properly. I wonder how the overexpressed HsGST exhibits its biological functions. The authors should explain this point.

6.In Figure 6, I think the control of pET-28a(+)-unrelated protein (for example BSA or others) is lacking.

7.In Figure 6, the authors did not mention the concentration of hydrogen peroxide used.

8.In Figure 6, can the authors compare the oxidative stress-protective effects of HsGST with other GSTs?

9.In line 208, the authors should mention who isolated the strain ANT108.

10.In line 272, the authors have mentioned that rHsGST may have the potential application as functional foods and therapeutic agents. However, there are already numerous GSTs available in the market. The authors need to specify this point.

Author Response

Response to Reviewer 1 Comments

Basically, this manuscript provides some fundamental data with novelty. However, there are some concerns needed to be addressed before publication.

Point 1: In Figure 3, I recommend the authors to combine Figure 3A and Figure 3B to on graph.

Response 1: We appreciate very much for the Reviewer’s good comments and kind recommendation. This part has been combined in this version, please see page 6 Figure 3.

Point 2: In Figure 3B lane 4, the band for rHsGST is not obvious. In addition, the authors did not explain the molecular weight and purity of rHsGST in the text.

Response2: We appreciate very much for the Reviewer’s comment. This part was revised, please see page 5 line 129-132 and page 6 Figure 3.

Point 3: The authors did not mention how to refold rHsGST.

Response 3: We appreciate very much for the Reviewer’s comment. This part was added, please see page 10 line 244-247.

Point 4: In lines 163-164, this sentence seems not related to the topic discussed.

Response: We appreciate very much for the Reviewer’s useful suggestion. This part has been deleted.

Point 5: In Figure 6, the authors overexpressed HsGST in BL21. Generally, the overexpressed proteins within E. coli are in the form of inclusion bodies and do not refold properly. I wonder how the overexpressed HsGST exhibits its biological functions. The authors should explain this point.

Response 5: We appreciate very much for the Reviewer’s good comments and kind recommendation. It’s normally that the overexpressed proteins within E.coli are in the form of inclusion bodies. In this study, the treatment of the inclusion bodies of HsGST were added (please see page 10 line 244-247), and the enzyme activity of the purified rHsGST were also detected. This method has been widely used in the studies of our laboratory [Wang, Y. et al, Marine Drugs 2018 16(10); Wang, Y. et al, Int. J. Biol. Macromol. 2018] and other reported studies [Sandamalika, W.M.G et al, Fish Shellfish Immun. 2018, 77, 252-263; Zhang, Y.Y. et al, Cell Stress Chaperon. 2013, 18, 503-51.].

Point 6: In Figure 6, I think the control of pET-28a(+)-unrelated protein (for example BSA or others) is lacking.

Response 6: In this study, BL21/pET-28(+) was set as the control, which accorded with the principle of single factor experiment design and could explain the functions of HsGST.

Point 7: In Figure 6, the authors did not mention the concentration of hydrogen peroxide used.

Response 7: We appreciate very much for the Reviewer’s comment. This part was revised, please see page 11 line 280.

Point 8: In Figure 6, can the authors compare the oxidative stress-protective effects of HsGST with other GSTs?

Response 8: We highly agree and appreciate very much for the Reviewer’s nice comments. The comparisons of the GSTs have been revised, please see page 9 line 206-207.

Point 9: In line 208, the authors should mention who isolated the strain ANT108.

Response 9: We appreciate very much for the Reviewer’s useful suggestion. This sentence was revised in this version, please see page 9 line 220-221.

Point 10: In line 272, the authors have mentioned that rHsGST may have the potential application as functional foods and therapeutic agents. However, there are already numerous GSTs available in the market. The authors need to specify this point.

Response 10: We appreciate very much for the Reviewer’s good comments and kind recommendation. The potential application of rHsGST has been added, please see page 11 line 289-291.

Reviewer 2 Report

The manuscript by Hou et al. described a new Glutathione S-transferase from Antarctic bacterium Halomonas sp. ANT108 possessing cold-adapted properties. This enzyme could be useful in several industries. In general, the results are interesting; however, the reviewer will encourage the authors to rearrange some part of the text so it becomes more reader friendly.

Please find my comments below:

Major comments

-          Please provide more information in the introduction about Halomonas which is the HSGST is originated from. Also, the first time the hsgst gene is referred to (page 2 line 55), novel hsgst gene was isolated from Antarctic psychrophilic bacterium ‘Halomonas …..’ should be added.

-          Please make a fair comparison and not only choose the selected ones that benefit the results. For example on page 2, line 64: ‘However, gst genes from other origins are longer than the HsGST gene’. Comparing 200 amino acid to 205 or 216 (later on EcGST, 202) is not significantly shorter. Also on page 4 line 119-120 ‘the specific activity of rHsGST was 254.20 nmol/min/mg. The data was higher than GST isolated from Liposcelis entomophila (183 nmol/min/mg)’ (It will be better to include other GSTs and not only the ones that will advantage the authors’ discussion). The authors use several GST sequences from other GSTs, but did not mention anything about them with respect to properties (and not mention for the comparison in line 65-66.

-          Figure 1 is too small and very low resolution.

-          It may be an idea to explain more on H- and G-sites (line 67-72) before mention about them.

-          Page 2, line 74-75: ‘HsGST 74 exhibited the highest overall levels of identity (90.0 to 32.0%) with other GSTs’. Please modify this sentence because it makes no sense.

-          In section 2.2, instead of mention ‘superimposed well (line 84-85)’, please consider providing other parameters, e.g. RMSD or sequence similarity/identity. Also, include the validation/quality parameters for model.

-          Figure 3, it seems to the reviewer that Figure 3A is not necessary because it is the same as Fig. 3B (lane 2 and 3). Could the authors explain why the purified protein is so low in concentration? It looks like it is pure, however it could also be that the authors purposely loaded the protein in lower concentration so the other impurity could not be visualized on the gel.  How much purified protein did the authors obtain from one production (from which scale, e.g. 1 L)?

-          Figure 4 is too small (or at least please enlarge the text size).

-          Figure 6 can be bigger especially the plates.

Minor comments

-          Page 1, line 13: please change bacteria to bacterium

-          What’s more should be changed to Furthermore

-          Page 6, line 151: please change ‘Meanwhile’ (that sentence does not make sense).

-          It may be an idea to use the same style for amino acids throughout the manuscript.

-          Considering changing CH3COONa/CH3COOH to acetate buffer.

Author Response

Response to Reviewer 2 Comments

The manuscript by Hou et al. described a new Glutathione S-transferase from Antarctic bacterium Halomonas sp. ANT108 possessing cold-adapted properties. This enzyme could be useful in several industries. In general, the results are interesting; however, the reviewer will encourage the authors to rearrange some part of the text so it becomes more reader friendly.

Please find my comments below:

Point 1: Please provide more information in the introduction about Halomonas which is the HSGST is originated from. Also, the first time the hsgst gene is referred to (page 2 line 55), novel hsgst gene was isolated from Antarctic psychrophilic bacterium ‘Halomonas …..’ should be added.

Response 1: We appreciate very much for the Reviewer’s comment. These two parts were revised, please see page 2 line 59-64.

Point 2: Please make a fair comparison and not only choose the selected ones that benefit the results. For example on page 2, line 64: ‘However, gst genes from other origins are longer than the HsGST gene’. Comparing 200 amino acid to 205 or 216 (later on EcGST, 202) is not significantly shorter. Also on page 4 line 119-120 ‘the specific activity of rHsGST was 254.20 nmol/min/mg. The data was higher than GST isolated from Liposcelis entomophila (183 nmol/min/mg)’ (It will be better to include other GSTs and not only the ones that will advantage the authors’ discussion). The authors use several GST sequences from other GSTs, but did not mention anything about them with respect to properties (and not mention for the comparison in line 65-66.

Response 2: We highly appreciated the Reviewer’s useful suggestion. All of them have been rectified in the manuscript, please see page 2 line 71-73 and page 5 line 133-135.

Point 3: Figure 1 is too small and very low resolution.

Response 3: We appreciate very much for the Reviewer’s good comments and kind recommendation. Figure 1 has been replaced, please see page 3 Figure 1.

Point 4: It may be an idea to explain more on H- and G-sites (line 67-72) before mention about them.

Response 4: We appreciate very much for the Reviewer’s comment. The explanations of H- and G-sites have been modified in introduction, please see page 2 line 78-83.

Point 5: Page 2, line 74-75: ‘HsGST 74 exhibited the highest overall levels of identity (90.0 to 32.0%) with other GSTs’. Please modify this sentence because it makes no sense.

Response 5: We appreciate very much for the Reviewer’s good comments and kind recommendation. This part has been revised, please see page 2 line 76-77.

Point 6: In section 2.2, instead of mention ‘superimposed well (line 84-85)’, please consider providing other parameters, e.g. RMSD or sequence similarity/identity. Also, include the validation/quality parameters for model.

Response 6: We appreciate very much for the Reviewer’s useful suggestion. The parameters of the model for homology modeling were added, please see page 3-4 line 96-100 and line 100-102.

Point 7: Figure 3, it seems to the reviewer that Figure 3A is not necessary because it is the same as Fig. 3B (lane 2 and 3). Could the authors explain why the purified protein is so low in concentration? It looks like it is pure, however it could also be that the authors purposely loaded the protein in lower concentration so the other impurity could not be visualized on the gel. How much purified protein did the authors obtain from one production (from which scale, e.g. 1 L)?

Response 7: We highly agree and appreciate very much for the Reviewer’s nice comments. The fermentation volume of the protein purification was added, please see page 10 line 243. The protein amount was also added, please see page 5 line 132. The target protein purified using Ni-NTA exhibited the high purity, and Figure 3 in page 6 was replaced in new version.

Point 8: Figure 4 is too small (or at least please enlarge the text size).

Response 8: We appreciate very much for the Reviewer’s good comments and kind recommendation. This part has been revised, please see page 7 Figure 4.

Point 9: Figure 6 can be bigger especially the plates.

Response 9: We highly appreciated the Reviewer’s useful suggestion. All of them have been rectified in the manuscript, please see page 9 Figure 6.

Point 10: Minor comments

- Page 1, line 13: please change bacteria to bacterium

- What’s more should be changed to Furthermore

- Page 6, line 151: please change ‘Meanwhile’ (that sentence does not make sense).

- It may be an idea to use the same style for amino acids throughout the manuscript.

- Considering changing CH3COONa/CH3COOH to acetate buffer.

Response 10: We highly appreciated the Reviewer’s useful recommendation. All of them have been rectified in the manuscript, please see page 1 line 13; page 5 line 117; page 7 line 163; page 2 line 78-85, page 4 line 106 and Table 1, page 5 Figure 2; page 10 line 260.

Reviewer 3 Report

The authors of the presented manuscript characterized a novel cold-adapted glutathione S-transferase, which was found in a bacterium isolated in the antarctic. In general, the research was carried out thouroughly and I am not lacking any experiments. However, especially the methods section needs some improvement to enable reproducibility. I would like to encourage the authors to consider the following comments:

p1, l29: Calling antarctica the worst environment on earth is a bit slang. It would be better to write "one of the most extreme environments".

p3, l84: replace beta-folds with beta-strands

p7, l180-181: Not the substrate has a high affinity for the enzyme, the enzyme has a higher affinity for GSH than for CDNB!

p8, l209-210: as you write this sentence it sounds like the strain was identified based on the growth temperature. please re-write.

p8, l210-211: E. coli BL21 or E. coli BL21 (DE3)?

p8, l222: How exactly was the homology modeling done? Can you really do it with Pymol or was Pymol just used for visualization? Which Pymol version was used?

p8, l228-229; l236: What is DE3? What is Kana? What is CDNB and GSH? I have guesses, but please specify abbreviations the first time you mention them.

p9, l253-254: What was the concentration of the other substrate respectively? Which buffer and pH was used?

p9, l255: replace Kcat with kcat

Author Response

Response to Reviewer 3 Comments

The authors of the presented manuscript characterized a novel cold-adapted glutathione S-transferase, which was found in a bacterium isolated in the antarctic. In general, the research was carried out thouroughly and I am not lacking any experiments. However, especially the methods section needs some improvement to enable reproducibility. I would like to encourage the authors to consider the following comments:

Point 1: p1, l29: Calling antarctica the worst environment on earth is a bit slang. It would be better to write "one of the most extreme environments".

Response 1: We highly appreciated the Reviewer’s useful recommendation. This part has been revised, please see page 1 line 29.

Point 2: p3, l84: replace beta-folds with beta-strands

Response 2: We appreciate very much for the Reviewer’s good comments and kind recommendation. This part has been modified, please see page 3 line 95.

Point 3: p7, l180-181: Not the substrate has a high affinity for the enzyme, the enzyme has a higher affinity for GSH than for CDNB!

Response 3: We highly agree and appreciate very much for the Reviewer’s nice comments. This sentence has been rectified, please see page 8 line 190-191.

Point 4: p8, l209-210: as you write this sentence it sounds like the strain was identified based on the growth temperature. please re-write.

Response 4: We appreciate very much for the Reviewer’s good comments and kind recommendation. This part has been revised, please see page 9 line 220-221.

Point 5: p8, l210-211: E. coli BL21 or E. coli BL21 (DE3)?

Response 5: We appreciate very much for the Reviewer’s comment. This part was modified, please see page 9 line 222.

Point 6: p8, l222: How exactly was the homology modeling done? Can you really do it with Pymol or was Pymol just used for visualization? Which Pymol version was used?

Response 6: We highly agree and appreciate very much for the Reviewer’s nice comments. The specific description of the homology modeling has been revised, please see page 10 line 234-235.

Point 7: p8, l228-229; l236: What is DE3? What is Kana? What is CDNB and GSH? I have guesses, but please specify abbreviations the first time you mention them.

Response 7: We appreciate very much for the Reviewer’s useful suggestion. All of the specify abbreviations in the whole manuscript were modified the first time we mentioned them, please see page 5 line 130, page 10 line 242, page 8 line 187-188 and page 1 line 38.

Point 8: p9, l253-254: What was the concentration of the other substrate respectively? Which buffer and pH was used?

Response 8: We appreciate very much for the Reviewer’s comment. The concentrations of the two substrates, and the buffer and pH used were all supplemented, please see page 11 line 269-271.

Point 9: p9, l255: replace Kcat with kcat

Response 9: This part has been replaced, please see page 8 Table 3 and page 11 line 272-273.

Reviewer 4 Report

The manuscript by Yanhua Hou and coworkers deals with the characterization of a novel glutathione S-transferases from an Antarctic Halomonas isolate. Even if the experimental design is clear and appropriate, in my opinion the manuscript should be improved.

- Introduction should be focused on the available information on GSTs from cold-adapted bacteria. I find it too general.

- Conclusions should be improved. In fact, it seems a summary of main results. Instead, I suggest to highlight the biotechnological potentialities of the analysed enzymes, based on results obtained in this study.

- In the materials and methods, please specified how the strain was isolated (which medium? Which temperature?? What about the incubatio duration?). Has the strain an accession number? Please, release the sequence.

- English should be improved.

- Please, avoid acronims in the abstract.

- line 130, "are shown"

- Please, avoid the use of What's more (lines 56-102-150)

In my opinion, the manuscript should be accepted for publication after major revisions.

Author Response

Response to Reviewer 4 Comments

The manuscript by Yanhua Hou and coworkers deals with the characterization of a novel glutathione S-transferases from an Antarctic Halomonas isolate. Even if the experimental design is clear and appropriate, in my opinion the manuscript should be improved.

Point 1: Introduction should be focused on the available information on GSTs from cold-adapted bacteria. I find it too general.

Response 1: We appreciate very much for the Reviewer’s good comments and kind recommendation. This part has been added, please see page 1 line 40-43.

Point 2: Conclusions should be improved. In fact, it seems a summary of main results. Instead, I suggest to highlight the biotechnological potentialities of the analyzed enzymes, based on results obtained in this study.

Response 2: We appreciate very much for the Reviewer’s useful suggestion. The whole conclusion was modified in this version, please see page 11 line 289-291.

Point 3: In the materials and methods, please specified how the strain was isolated (which medium? Which temperature?? What about the incubatio duration?). Has the strain an accession number? Please, release the sequence.

Response 3: We highly agree and appreciate very much for the Reviewer’s nice comments. This part has been revised, please see page 9 line 217-222.

Point 4: English should be improved.

Response 4: We appreciate very much for the Reviewer’s useful suggestion. In the whole manuscript, those English language mistakes have been rectified in Blue.

Point 5: Please, avoid acronims in the abstract.

Response 5: We appreciate very much for the Reviewer’s good comments and kind recommendation. The acronims in the abstract has been deleted. Please see the whole abstract in page 1 line 11, 13-14, 17 and 21.

Point 6: line 130, "are shown"

Response 6: This part has been rectified in page 6 line 141.

Point7: Please, avoid the use of What's more (lines 56-102-150)

Response 7: We highly agree and appreciate very much for the Reviewer’s nice comments. Those “What’s more” in the whole manuscript have been revised, please see page 2 line 51, 54, page 5 line 117 and page 7 line 161.

In my opinion, the manuscript should be accepted for publication after major revisions.

Round 2

Reviewer 1 Report

The method of refolding rHsGST is still missing. The authors should add this part.

In lines 288-289, "These special catalytic properties indicated that rHsGST might have the potential application as functional foods and therapeutic agents.", this sentence is recommended to be deleted.

Author Response

Response to Reviewer 1 Comments

Point 1: The method of refolding rHsGST is still missing. The authors should add this part.

Response 1: We appreciate very much for the Reviewer’s good comments and kind recommendation. This part has been rectified in this version, please see page 10 line 247-249.

Point 2: In lines 288-289, "These special catalytic properties indicated that rHsGST might have the potential application as functional foods and therapeutic agents.", this sentence is recommended to be deleted.

Response2: We appreciate very much for the Reviewer’s comment. This part has been revised in this version, please see page 11 line 290.
